# AN INTRINSIC DIMENSION PERSPECTIVE OF TRANSFORMERS FOR SEQUENTIAL MODELING

## ABSTRACT

Transformers have become immensely popular for sequential modeling, particularly in domains like natural language processing (NLP). Recent innovations have introduced various architectures based on the Transformer framework, resulting in significant advancements in applications. However, the underlying mechanics of these architectures are still somewhat enigmatic. In this study, we explore the geometrical characteristics of data representations learned by Transformers using a mathematical metric known as intrinsic dimension (ID). This can be conceptualized as the minimum parameter count needed for effective modeling. A sequence of experiments, predominantly centered on text classification, support the ensuing empirical observations regarding the correlation between embedding dimension, layer depth, individual layer ID, and task performance. Interestingly, we note that a higher terminal feature ID, when obtained from Transformers, generally correlates with a lower classification error rate. This stands in contrast to the behavior observed in CNNs (and other models) during image classification tasks. Furthermore, our data suggests that the ID for each layer tends to diminish as layer depth increases, with this decline being notably steeper in more intricate architectures. We also present numerical evidence highlighting the geometrical constructs of data representations as interpreted by Transformers, indicating that only nonlinear dimension reduction is achievable. Lastly, we delve into how varying sequence lengths impact both ID and task performance, confirming the efficacy of data reduction during training. Our ambition is for these insights to offer direction in the choice of hyper-parameters and the application of dimension/data reduction when using Transformers for text classification and other prevalent NLP tasks.

## 1 INTRODUCTION

Transformers, as introduced by (Vaswani et al., 2017), have revolutionized numerous machine learning disciplines. They've notably driven breakthroughs in both natural language processing (NLP) and computer vision (CV). The Transformer architecture excels at managing vast datasets, predominantly when equipped with an ample number of parameters. Prominent models like BERT (Devlin et al. (2018)), GPT-3 (Brown et al. (2020)), and BART (Lewis et al. (2019)) exemplify its prowess. Impressively, given sufficient training data, Transformers often surpass rivals like CNNs in performance (Dosovitskiy et al. (2020)).

As researchers continue to unlock the potential of Transformers, a plethora of variations have surfaced. The Reformer (Kitaev et al. (2020)), for instance, mitigates the computational complexity from $O(L^2)$ to $O(L \log L)$ using locality-sensitive hashing, where $L$ represents the sequence length. The Sparse Transformer (Child et al. (2019)) employs sparse factorizations, achieving a similar reduction in memory overhead. Linformer (Wang et al. (2020)) adopts low-rank matrices to approximate the self-attention mechanism, decreasing both computation and memory costs from $O(L^2)$ to $O(L \log L)$.

Furthermore, DRformer (Tang & Huang (2022)) is adept at recognizing both regional and attribute relations within the pedestrian attribute recognition domain. The Tightly-coupled convolutional transformer (Shen & Wang (2022)) enhances the granularity of information retrieval in time series forecasting. ACORT (Tan et al. (2022)) streamlines the Transformer's footprint for image captioning

tasks. Lastly, Huang et al. (2022) has designed a Transformer variant tailored for news recommendation.

While there has been remarkable progress in architecture development, the foundational mechanics behind Transformers remain elusive. Training a Transformer is notoriously challenging, and we've yet to fully understand its inner workings, especially regarding how performance varies with increasing embedding dimension and depth. Yet, understanding these nuances is vital, as there's a trend towards designing bigger, deeper Transformers with enhanced training techniques to bolster performance. Some recent studies have begun to shed light on this. Xiong et al. (2020) and Popel & Bojar (2018) explored the impact of hyper-parameter tuning on Transformer training. Huang et al. (2020) examined the complexities of optimizing Transformer models and introduced a novel initialization approach for training deeper iterations. Efforts to train expansive Transformer models were also seen in Wang et al. (2019), with Wang et al. (2022) successfully training a Transformer boasting a depth of 1000 layers. This paper primarily endeavors to demystify the Transformer's operational intricacies by focusing on its intrinsic dimension (ID) representation.

It's broadly acknowledged that real-world data, encompassing sounds, texts, images, and the like, inherently exhibit low-dimensional structures. This implies that only a subset of dimensions is necessary to encapsulate sampled data and inherent relationships.

Consider, for instance, the dataset comprising all $224 \times 224 \times 3$ RGB images labeled as 'dogs'. While there exist $224 \times 224 \times 256 \times 3 = 38535168$ possible configurations, the actual, "intrinsic" count of distinct dog images perceivable by humans is substantially fewer, with many bearing notable resemblances. Numerous deep learning strategies, such as Hinton & Salakhutdinov (2006) and Gonzalez & Balajewicz (2018), leverage the prevalence of these low-dimensional data constructs.

The concept of intrinsic dimension (ID), as described in Amsaleg et al. (2015), Houle et al. (2012), Cutler (1993), and Xie et al. (2009), serves as a critical mathematical instrument to fathom the geometric essence of data. It signifies the minimal parameter set necessary to model specific truths, ideally pinpointing the low-dimensional data architectures. Our study mainly probes the data representation by Transformers, striving to highlight unique phenomena through the lens of ID.

Our work's salient contributions are fourfold:

- We examine the ID fluctuation of data representations as deciphered by sequential Transformer blocks, identifying a prominent *dimension reduction* trend across layers. This trend is accentuated with deeper architectures.

- We illuminate the geometric nuances of Transformers for sequential modeling. Notably, it appears Transformers primarily achieve nonlinear dimension reduction in text classification tasks.

- We delve into the interplay between embedding dimension (ED), the intrinsic dimension (ID) of the deduced representation, and task performance. This provides a clear rationale behind the advantages of expanding ED from an ID viewpoint.

- We assess the implications of truncating training datasets through sequence lengths, noting minimal impact on both IDs and performance metrics. This insight might offer pragmatic guidance for refining computational efficiency in real-world scenarios.

## 2 RELATED WORK

**TwoNN Method** There has been extensive research on estimating the intrinsic dimension (ID) of provided datasets. For instance, methods based on Principal Component Analysis (PCA) can be found in Fukunaga & Olsen (1971) and Bruske & Sommer (1998). The maximum likelihood estimation (MLE) based approach is described in Levina & Bickel (2004). Costa & Hero (2004) leveraged the geodesic-minimal-spanning-tree for ID estimation, and Kégl (2002) employed the capacity dimension for the same purpose. An innovative estimation method, named TwoNN, is presented in Facco et al. (2017). This method uses the proximity of the nearest and the second nearest samples to shape probability distributions. Due to its computational efficiency, we have

chosen the TwoNN method for estimating the intrinsic dimensions of the representations decoded by Transformers.[1]

Formally, the TwoNN method operates as follows. Given the dataset $\mathcal{D} = \{x_i\}_{i=1}^N$, for each data point $x_i$, let the distances to its closest and second closest samples be denoted by $s_{i,1}$ and $s_{i,2}$, respectively. The intrinsic dimension ID is estimated by analyzing the ratio:

$$s_i := \frac{s_{i,2}}{s_{i,1}}.$$

Notably, the ratio $s_i := \frac{s_{i,2}}{s_{i,1}}$ can be characterized by a Pareto distribution, as detailed in Hussain et al. (2018) and Rootzén & Tajvidi (2006). This gives rise to the equation:

$$P((s_1, s_2, ..., s_N) \mid d) = d^N \prod_{i=1}^N s_i^{-(d+1)}.$$

The intrinsic dimension of the dataset $\mathcal{D}$, denoted by $d$, can then be efficiently determined using a standard maximum likelihood technique. For clarity, a step-by-step outline of the TwoNN method is provided below:

- Randomly select $N$ data points, forming $\mathcal{D} = \{x_i\}_{i=1}^N$.
- For each $x_i$ in $\mathcal{D}$, compute the distances to the closest and second closest data points, labeled as $s_{i,1}$ and $s_{i,2}$.
- For every $x_i$ in $\mathcal{D}$, determine the ratio $s_i := \frac{s_{i,2}}{s_{i,1}}$.
- Estimate $d$ from

$$P((s_1, s_2, ..., s_N) \mid d) = d^N \prod_{i=1}^N s_i^{-(d+1)}$$

  using the maximum likelihood technique. This $d$ serves as the ID estimate.

**Intrinsic Dimension in Deep Learning**  There has been recent interest in examining the role of intrinsic dimension within deep learning frameworks. For instance, Pope et al. (2021) employed GANs to generate synthetic images, allowing them to establish an upper bound for ID, with the goal of verifying the precision of ID estimation methodologies. Ansuini et al. (2019) explored variations in ID across layers for traditional neural network architectures, including ResNet (He et al. (2016)) and VGG (Simonyan & Zisserman (2014)), focusing on image classification tasks. Aghajanyan et al. (2020) delved into the impact of pre-training on the intrinsic dimension in the context of NLP tasks. However, when juxtaposed with CNNs and other models tailored for computer vision, there is a paucity of research concerning sequential models designed for NLP, such as Transformers, despite their widespread acclaim. This study endeavors to bridge this research void.

## 3 RESULTS

We commence by detailing our experimental setup and subsequently present our findings across five dimensions.

### 3.1 EXPERIMENTAL SETTINGS

**Tasks**  In order to conveniently examine the intrinsic dimension (ID) of Transformers, we have chosen to focus our numerical experiments on the text classification task. Our rationale is three-fold. Firstly, text classification stands as a foundational yet crucial task within the realm of natural language processing. Its utility spans a broad array of applications, including but not limited to, spam detection (Crawford et al. (2015), Asghar et al. (2020)), text style categorization (Wu et al. (2019), Sudhakar et al. (2019)), and sentiment analysis (Medhat et al. (2014), Xu et al. (2019)). Secondly, numerous studies, such as Devlin et al. (2018), Wang et al. (2020), and Shaheen et al. (2020), have underscored the effectiveness of the Transformer architecture in classification tasks.

---

[1]Refer to `https://github.com/ansuini/IntrinsicDimDeep` for relevant code.

Lastly, for text classification purposes, the Transformer typically employs a sequence of encoder blocks, augmented with a supplementary multi-layer perceptron (MLP) (Rosenblatt (1961)) block for classification. This streamlined approach facilitates the ease of ID analysis. Conversely, for tasks necessitating decoders, such as text generation, dissecting ID on a layer-by-layer basis can pose challenges.

**Datasets** We conducted our experiments on three distinct datasets: IMDB (Maas et al. (2011)), AG (Zhang et al. (2015)), and SST2 (Socher et al. (2013)). Specifically, IMDB (Maas et al. (2011)) is a binary classification dataset for movie reviews, comprising 25,000 training entries and 25,000 test entries. The AG dataset (Zhang et al. (2015)) consists of news articles categorized into four classes, with 120,000 training entries and 7,600 test entries. Meanwhile, SST2 is another binary movie review classification dataset, housing approximately 7,000 training entries and 2,000 test entries.

**Models** We employ the standard Transformer model (Vaswani et al. (2017)), leveraging successive encoder blocks to extract features and represent the entirety of the input texts. For the final classification layer, in line with common practices and for the sake of simplicity, we utilize standalone multi-layer perceptrons (MLPs) (Rosenblatt (1961)). These maintain a significant influence on the Transformer's (encoders) overall performance.

**Objectives and Relevant Hyper-parameters** This study seeks to understand and elucidate the correlation between the intrinsic dimensions of representations derived by Transformers and the ensuing classification outcomes. In striving for a thorough and robust experimental approach, we modulated the following hyper-parameters:

- $D$: Depth, representing the total layer count within the Transformer.
- ED: Embedding (Mikolov et al. (2013)) dimension. For clarity and in adherence to established conventions (as illustrated by Vaswani et al. (2017)), we maintained a consistent dimension across all hidden layers.

Essentially, our aim is to discern their impacts on both the ID and classification accuracy.

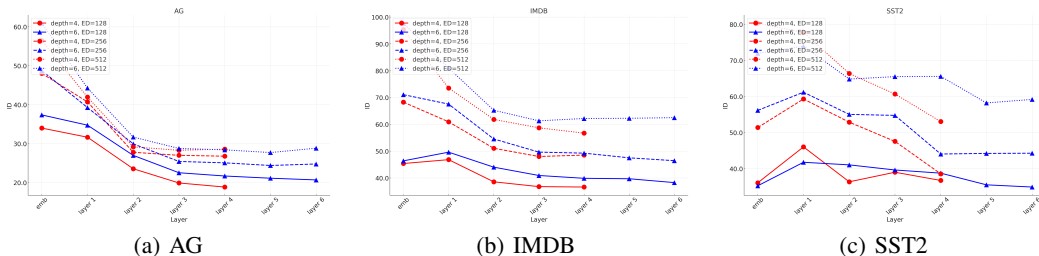

Figure 1: Layer-wise Variation of Intrinsic Dimensions across Different Depths and Embedding Sizes

## 3.2 LAYER-WISE VARIATION OF INTRINSIC DIMENSIONS

We commence by examining the ID variation across different layers. In a Transformer model of depth $D$ and hidden dimensions $\{n_l\}_{l=1}^{D}$, every piece of input data is sequentially mapped into a $n_l$-dimensional vector space for $l = 1, 2, \cdots, D$. Despite the defined hidden dimensions, these cannot truly depict the underlying geometric structures of the data. Thus, we employ the TwoNN (Facco et al. (2017)) method to ascertain the ID for each layer.

Building upon the settings delineated in Section 3.1, our experiments span three datasets (AG, IMDB, and SST2) using the Transformer model with various depths and embedding dimensions: $D \in \{4, 6\}$ and ED $\in \{128, 256, 512\}$. The intrinsic dimension and its layer-wise changes are illustrated in Figure 1.

In Figure 1, the horizontal axis indicates each layer, with $\mathrm{emb}$ symbolizing the embedding layer and layer $i$ referring to the $i$-th encoder layer. The vertical axis quantifies the corresponding ID, and each line in Figure 1 charts the ID's layer-wise variations for a specific hyper-parameters setup.

Key observations from Figure 1 include:

- As depth increases, the overall intrinsic dimension generally displays a declining trend. Typically, the ID may surge at the first encoder layer (refer to Figure 1 (c)), yet it diminishes in subsequent layers, bottoming out at the final layer. This decrement in ID across layers can be perceived as a form of *dimension reduction* in Transformers, corresponding with the distillation of pertinent information for the concluding classification.

- Keeping the model's depth constant for a given dataset reveals that the intrinsic dimension tends to escalate with the embedding dimension. As corroborated by Figure 1, this observation persists across all depths and datasets. This trend is intuitive; as the embedding dimension grows, the starting intrinsic dimension generally follows suit, culminating in a higher ending ID.

To delve deeper into this dimension reduction phenomenon, we can inspect the variation in the ratios between the intrinsic dimension and both the embedding and hidden dimensions. Given that the latter two were predefined as equivalent, this analysis becomes straightforward. Notably, the ratio starts around $O(10^{-1})$ at the embedding layer (roughly between 0.1 and 0.3), but significantly drops to a value near $O(10^{-2})$ by the final layer.

## 3.3 RELATIONSHIP BETWEEN INTRINSIC DIMENSIONS AND CLASSIFICATION ACCURACY

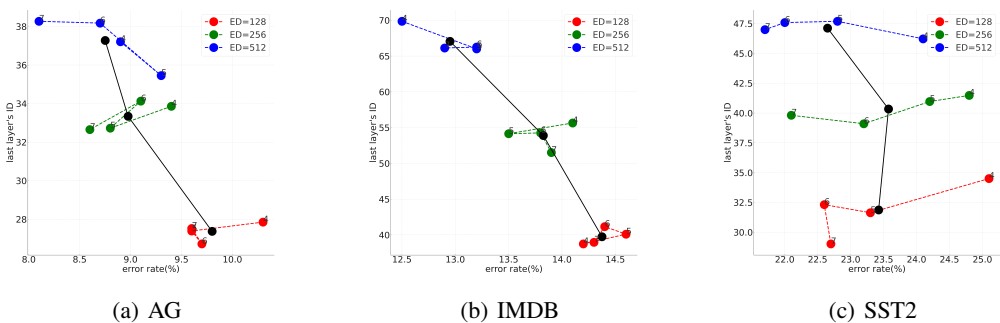

(a) AG         (b) IMDB         (c) SST2

Figure 2: Relationship Between Terminal Intrinsic Dimension and Classification Error Rate Across Datasets

Given that the ID describes the inherent geometric structures of data distribution, and that classification success hinges on the final representation produced by the Transformers' last hidden layer, one could posit that the terminal ID and predicted classification accuracy are related.

To further explore this, we examined the relationship between terminal ID and its corresponding classification error rate across different hyper-parameter settings and datasets, as depicted in Figure 2. For consistency, training was done through multiple independent runs with random initializations to ensure convergence. In Figure 2, the x-axis displays the classification error rate, while the y-axis presents the ID of the representation derived from the final hidden layer. Dashed lines link outcomes for various depths at a set embedding dimension, with black points indicating average values for both terminal ID and error rates.

A glance at Figure 2 reveals that, despite all tests being conducted within the same hypothesis space (namely, Transformers), notable discrepancies exist in terms of classification efficacy and terminal ID in relation to model size (or embedding dimension in this context). As illustrated by the solid lines in Figure 2, the terminal ID generally rises with increasing embedding dimension, but classification error displays the opposite trend. Fascinatingly, the trend seen for Transformers in NLP contrasts

sharply with findings in Ansuini et al. (2019), where the terminal ID for CNNs directly correlates with the classification error rate for image processing.

The rationale behind this can be understood as follows. As the embedding dimension increases, in line with the observations in Section 3.2, there's an increase in the ID of the input data representation, leading to a rise in the terminal ID. Simultaneously, a larger embedding dimension, along with greater hidden dimensions, certainly expands the associated hypothesis space for modeling. Therefore, it's logical to expect an improvement in classification accuracy.

This observation suggests that boosting the embedding dimension can elevate classification performance by amplifying the intrinsic dimension.

This insight paves the way for potential practical guidelines in applications: one might conceptualize a methodical approach to leverage the terminal ID as a *posterior* "indicator" of generalization. Specifically, by tracking the changes in terminal ID throughout training, we might be able to ensure superior classification outcomes without needing an entirely separate dataset for both validation and testing. This approach could be especially beneficial when dealing with scarce data in real-world scenarios, making it a promising avenue for future research.

**Remark 1** *Our analysis underscores the intrinsic behavior of transformer models in textual tasks. However, for ViT models, the findings deviate, as evidenced in Table 1. We observe that both the ViT embedding dimension and the ViT depth have minimal impact on the ID.*

Table 1: The correlations between the ID and classification accuracy of ViT model on CIFAR-10 dataset. The left subtable shows the results under a fixed depth and the right one shows those under a fixed embedding dimension.

| fixed depth=7 | last layer's ID | fixed ED=384 | last layer's ID |
|---|---|---|---|
| ED=192 | 23.52 | depth=3 | 22.02 |
| ED=288 | 23.07 | depth=5 | 22.33 |
| ED=384 | 22.51 | depth=7 | 22.51 |

### 3.4    A PRINCIPAL COMPONENT ANALYSIS VIEWPOINT OF DATA REPRESENTATION

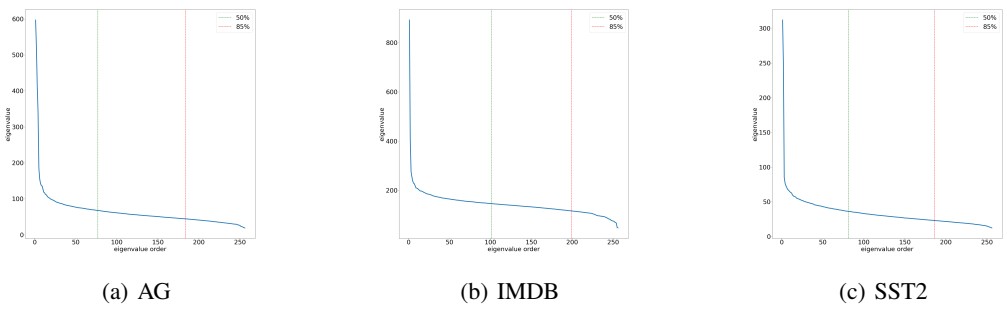

|     (a) AG     |     (b) IMDB     |     (c) SST2     |

Figure 3: PCA results on data representation of the last hidden layer (ED = 256, depth = 6). The vertical axis shows the value of corresponding eigenvalues. The green and red vertical dotted lines denote the number of components required to capture 50% and 85% of the variance in data representation.

There exist numerous tools for estimating the intrinsic dimension of data representation, including methods rooted in principal component analysis (PCA) (Fukunaga & Olsen (1971), Bruske & Sommer (1998)), as well as the TwoNN approach (Facco et al. (2017)), among others. Owing to its computational efficiency, we opted for the TwoNN algorithm. For a more in-depth discussion, readers can refer to Section 2.

Based on the findings presented in Figure 1, it's clear that the intrinsic dimensions of Transformers are significantly less than the embedding and hidden dimensions. In this section, we delve deeper,

demonstrating that the data representation crafted by Transformers resides on low-dimensional yet curved manifolds, rather than flat subspaces, rendering them unsuitable for model reduction through linear techniques.

To substantiate this claim, we employed the time-honored PCA method (Pearson (1901)) on the normalized covariance matrix of each Transformer layer across various hyper-parameter configurations and all three datasets. The outcomes for the final hidden layer are depicted in Figure 3. The x-axis ranks the eigenvalues of data representation in descending order, while the y-axis conveys the magnitude of these eigenvalues. The green and red dotted vertical lines represent the number of components necessary to account for 50% and 85% of data representation variance, respectively. The x-coordinate indicated by the red line is termed PCA-ID, representing the "pseudo" intrinsic dimension ascertained through a straightforward PCA technique.

The ID derived via the TwoNN methodology is substantially smaller than the PCA-ID. For instance, the ID presented in Figure 1 (a) is roughly 34 for ED = 256 on the AG dataset. In contrast, the PCA-ID displayed in Figure 3 (a) stands at about 180 in the same scenario, a figure that's 5-6 times greater. Moreover, the PCA-ID's ratio to the embedding dimension oscillates between 0.7 and 0.9, far exceeding that of the ID (ranging from 0.05 to 0.15). The stark disparity between the IDs inferred from the TwoNN and PCA methods highlights the pronounced non-linearity interlinking data samples. From these observations, we infer that the domain in which data representation is situated isn't a linear subspace but rather a distinct curved manifold. This nature precludes any attempts at rudimentary linear model reduction.

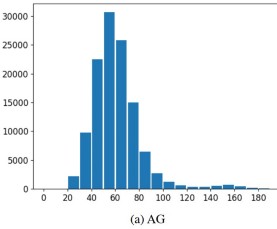 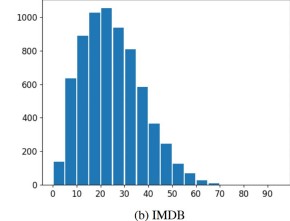 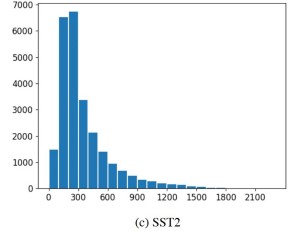

| (a) AG | (b) IMDB | (c) SST2 |

Figure 4: The length distribution histogram of several datasets. The horizontal axis represents sentence length, and the vertical axis represents frequency per length interval.

## 3.5 THE EFFECT OF INTRINSIC DIMENSION REDUCTION WITH RESPECT TO DEPTH

Table 2: The ID reduction w.r.t. depth for different embedding dimensions on the AG dataset, where "emb layer" denotes the embedding layer.

|  | depth=4 | | | depth=8 | | |
| --- | --- | --- | --- | --- | --- | --- |
|  | emb layer's ID | last layer's ID | decrease | emb layer's ID | last layer's ID | decrease |
| ED=128 | 42.04 | 27.85 | 14.19 | 43.81 | 27.14 | **16.66** |
| ED=256 | 54.91 | 33.86 | 21.05 | 55.09 | 32.44 | **22.65** |
| ED=512 | 60.93 | 37.21 | 23.73 | 60.96 | 32.46 | **28.50** |

Table 3: The ID reduction w.r.t. depth for different embedding dimensions on the IMDB dataset.

|  | depth=4 | | | depth=8 | | |
| --- | --- | --- | --- | --- | --- | --- |
|  | emb layer's ID | last layer's ID | decrease | emb layer's ID | last layer's ID | decrease |
| ED=128 | 52.36 | 38.74 | **13.62** | 51.14 | 37.68 | 13.46 |
| ED=256 | 73.91 | 55.65 | 18.27 | 72.98 | 47.30 | **25.68** |
| ED=512 | 102.41 | 69.86 | 32.55 | 103.33 | 60.55 | **42.78** |

In Transformers, as with other neural network architectures, model depth is a pivotal factor. A model that's too shallow might lack robust representational capability, leading to subpar training outcomes. Conversely, an exceedingly deep model might face generalization challenges like overfitting, resulting in diminished test performance, and might demand prohibitive computational and memory resources. In this section, we delve deeper into how ID variation correlates with model depth.

Table 4: The ID reduction w.r.t. depth for different embedding dimensions on the SST2 dataset.

| | depth=4 | | | depth=8 | | |
| | emb layer's ID | last layer's ID | decrease | emb layer's ID | last layer's ID | decrease |
|---|---|---|---|---|---|---|
| ED=128 | 40.85 | 34.50 | 6.34 | 41.39 | 30.12 | **11.26** |
| ED=256 | 52.29 | 41.48 | **10.82** | 48.32 | 37.87 | 10.45 |
| ED=512 | 60.30 | 46.22 | 14.08 | 59.8 | 43.27 | **16.53** |

As illustrated in Figure 1, there's a noticeable decline in ID across layers. To better understand the influence of model depth on this trend, it's instructive to examine the disparities between the IDs of embedding layers and those of the final hidden layers. Though the pertinent ID outcomes are already depicted in Figure 1, we've collated them in Table 2, 3, and 4 for clarity.[2] From this data, several key observations emerge:

- Holding the dataset and embedding dimension constant, the IDs of the embedding layers remain largely unchanged despite variations in depth.

- Again, with a consistent dataset and embedding dimension, the IDs of the final hidden layers typically diminish substantially as depth increases.

- Synthesizing the above insights reveals that Transformers exhibit a more pronounced dimension reduction across layers in deeper architectures.

A straightforward rationale for these observations can be posited. When the embedding dimension is held constant, the learned data representation tends to be stable, suggesting consistent IDs for the embedding layers. Simultaneously, referencing Figure 1 and the discourse in Section 3.2, the trend of decreasing ID seems to intensify in deeper models.

## 3.6 DATA REDUCTION CASES: IMPACT OF SEQUENTIAL LENGTHS

In the context of practical applications, the Transformer model is often associated with a considerable volume of parameters. This complexity necessitates the utilization of large datasets to achieve optimal training results, implicating notable space and computational time expenditures. The implication of these overheads motivates an investigation into the potential for reducing the size of the training dataset—a concept we refer to as "*data reduction*"—while maintaining, or potentially enhancing, the model's performance during testing.

Our study is anchored in the exploration of the feasibility and approaches to implementing data reduction in the training phase of Transformers. We approach this investigation by considering the intrinsic dimensionality of data representation. A notable observation from our preliminary findings is the relative stability of the intrinsic dimension, contrasting with the variability often observed in other hyper-parameters, such as the embedding dimension. This stability is observed irrespective of the sequential length of data samples, unveiling prospective avenues for data reduction through the selective exclusion of specific training samples without compromising the model's performance.

Our journey into this inquiry commences with an examination of the sequential length of samples constituting the training dataset. We systematically organize all training sentences according to their length, measured in terms of word count (refer to Figure 4 for a visual representation). A subset of sentences, termed the "long-set," is compiled by selecting the upper 80% of the most extended sentences present in the training dataset. Analogously, a "short-set" is constructed by harvesting the upper 80% of the shortest sentences. For comparative purposes, we retain the original, unmodified training dataset, denoting it as the "full-set."

The underpinning rationale for this categorization and selective inclusion is to mitigate the influence of outliers, particularly those that are exceptionally short or long. We posit that this strategy facilitates a more refined and coherent training dataset, thereby augmenting the integrity and efficacy of the model training process. It is of paramount importance to underscore that this modification is exclusive to the training datasets; the *test datasets remain unaltered*, ensuring the validity and integrity of our experimental results and conclusions.

---

[2]Due to space limitations, the depth 6 scenario is omitted.

This nuanced approach to data curation and reduction not only aims at optimizing computational efficiency but also seeks to unravel insights into the intricate dynamics between dataset size, intrinsic dimensionality, and model performance. Each subset—long-set, short-set, and full-set—serves as a distinct lens through which we scrutinize the interplay of these factors, laying a foundational framework for future research in efficient and effective training of Transformer models.

Table 5: The effect of sequential lengths on IDs of embedding layers and classification errors, where the depth of Transformers is fixed as 5.

|  |  | ID of embedding layer | | | error rate | | |
|---|---|---|---|---|---|---|---|
|  |  | full | long | short | full | long | short |
| AG | ED=128 | 43.77 | 46.23 | 44.35 | 9.6 | 10.3 | 10.6 |
|  | ED=256 | 55.49 | 54.07 | 57.21 | 8.8 | 9.5 | 10.1 |
|  | ED=512 | 58.96 | 59.22 | 59.12 | 9.3 | 9.2 | 9.4 |
| IMDB | ED=128 | 51.10 | 51.17 | 48.29 | 14.6 | 14.7 | 15.3 |
|  | ED=256 | 74.38 | 74.77 | 71.78 | 13.5 | 13.6 | 14.1 |
|  | ED=512 | 101.34 | 103.70 | 104.15 | 13.2 | 13.2 | 13.9 |
| SST2 | ED=128 | 42.10 | 41.63 | 42.16 | 23.3 | 25.9 | 26.1 |
|  | ED=256 | 53.17 | 53.14 | 54.40 | 24.2 | 26.2 | 25.5 |
|  | ED=512 | 62.23 | 60.51 | 62.84 | 22.8 | 24.6 | 24.3 |

### 3.7 EXPERIMENTAL ANALYSIS OF DATA REDUCTION BY SEQUENTIAL LENGTHS

Experiments were conducted using the three aforementioned training sub-datasets: long-set, short-set, and full-set. These experiments considered various configurations, with a primary focus on embedding dimensions, across different datasets. For every Transformer model trained within these parameters, we documented both the IDs of the embedding layers and the ultimate classification error rates (using the complete test datasets) in Table 5.

A principal observation emerges from Table 5. Reviewing the results row by row, we find that neither the IDs of the embedding layers nor the classification error rates show substantial variations, even though each Transformer model is trained using distinct subsets of the original training dataset. This finding lends support to the feasibility of data reduction in training, particularly with regard to sequential lengths. Such insights can prove invaluable in scenarios where data is scarce, highlighting avenues for further investigation in subsequent studies.

## 4 CONCLUSION

We have presented an analysis that offers a new perspective on understanding Transformers, emphasizing the role of intrinsic dimensions in data representation. This exploration has unveiled distinct patterns and relationships between intrinsic dimensions, task performance, and various hyperparameters including model depth, embedding dimensions, and data sequence length.

Our results highlight several core insights. We found a direct relationship between embedding dimensions, intrinsic dimensions, and classification accuracy. Additionally, we noted a decrease in intrinsic dimensions through successive layers, a trend amplified in models with increased depth.

In the context of the relationship between model and data, evidence supports that the data representation learned by Transformers is situated on curved manifolds. Our study also suggests that data reduction during the training phase can be effective, offering avenues for enhanced data efficiency and applicability in real-world scenarios.

Though our study is grounded in classification tasks, we are poised to expand this research into generative domains. We are also preparing to delve deeper into the examination of established architectures, including pre-trained language models, to enrich our understanding. Our future work is geared towards quantitative analyses that will connect the dots between Transformers and intrinsic dimensions, addressing existing theoretical voids.

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
