# OpenReview forum: "An Intrinsic Dimension Perspective of Transformers for Sequential Modeling"
_ICLR.cc/2024/Conference — ICLR 2024 Conference Withdrawn Submission_

### Official Review · Reviewer_h8wB · 2023-10-27

**Soundness:** 2 fair
**Presentation:** 3 good
**Contribution:** 2 fair
**Rating:** 5
**Confidence:** 4

**Summary:**

The manuscript describes the results of a series of empirical experiments on transformers, aimed at deciphering the structure of data representations their inner layer, with a focus on NLP datasets. In particular, the authors estimate the intrinsic dimension (ID) of the different representations, studying its variations across the layers, and as a function of the embedding dimension. It is found that the ID increases with the embedding dimension, and it decreases monotonically from the input to the output.  Moreover, it is  studied  the impact  of using  subsets of the training set on the ID and on the generalisation error.

**Strengths:**

Studying the internal representation of deep networks, and in particular of transformers, is a very timely and important research line, and the ID is one of the tools which allows addressing this task. The monotonic dependence of the ID on the depth is at odds with what observed in convolutional NN for image classification, and also with a similar analysis performed on transformers trained by self-supervision (https://arxiv.org/abs/2302.00294). Also the focus on the impact of sequence length on learning  (section 3.6 and 3.7) is interesting.

**Weaknesses:**

The observation that the ID grows with the embedding dimension  (ED), while the generalisation error decreases (bullet point number 3) seems to me pretty trivial. Of course by enlarging the ED one recovers a richer presentation, whose ID will be larger, and which will provide better models.
I was not able to understand if the ID is computed by performing a prior average pooling over the sequence, as done in other works on transformers. If this pooling is not performed, the dimension of the representation is equal to the number of tokens times ED, and not to ED.
The analysis performed in 3.6 is in principle interesting, but inconclusive (the results in table 5 seem to show that the ID cannot be used as a quality proxy to decide if the learning set can be reduced)
The analysis presented in 3.4 is very similar to the one presented in [Ansuini 2019]

**Questions:**

1) What is the space on which the ID is computed? The representation of the single tokens, or the representation of the sentence?
2) In ref https://arxiv.org/abs/2302.00294 it is observed a dependence of the ID on the depth which is non-monotonic. Even if the analysis is performed on different datasets, the monotonic behaviour seems to me at odds with the hypothesis that the first layer of a NN expand the dimension of the representation by getting rid of low-level confounders. Any explanation?
3) Can the ID really be used to guide the choice of the training set size?
4) What is  the take-home-message of the observation that the ID grows with the ED? Why this observation is not obvious as it seems?

---

### Official Review · Reviewer_LFkQ · 2023-10-29

**Soundness:** 2 fair
**Presentation:** 3 good
**Contribution:** 2 fair
**Rating:** 3
**Confidence:** 5

**Summary:**

Transformer has been a dominant neural network in natural language processing. The paper attempts to give insight analysis of Transformer from the viewpoint of intrinsic dimension analysis. Specially, the paper conducted experiments to explore the relationship between intrinsic dimension and performance of Transformers over classification tasks, as well as some important hyper-parameter in Transformers (such as embedding dimension, number of layers). The observation is that the intrinsic dimension will decrease through successive layers, and the higher classification accuracy generally corresponds to larger intrinsic dimension.

**Strengths:**

1.The understanding of Transformer is crucial in field of natural language process, the paper supplied a viewpoint from intrinsic dimension analysis to uncover the behavior of Transformer in sequential classification.

**Weaknesses:**

1.Several important related works on analysis of Transformers are missed (such as Revisiting over-smoothing in BERT from the perspective of graph), and the observations of decreasing intrinsic dimension can be viewed as a showcase of over-smoothing.
2.The experiments should be further improved since the paper resorts to experimental study. For example, more sequential learning tasks should be includes, only text classification seems insufficient, such as machine translation which is a typical sequential learning task for Transformer understanding (see more in question part).
3.Lack of deep understanding of the observation, for example, increasing the embedding dimension will increase (intrinsic dimension). However, large embedding dimension may suffer from over-fitting even with fixed depth (see more in question part).

**Questions:**

1.For experimental study, increasing depth/width of Transformer based encoder results in lower/higher ID. Do we have a trade-off between these two important hyper-parameters? The paper states that higher ID usually means lower error rate.  Does this mean that a shallow and wide structure is always better than deep structure for text classification?

2.Comparing to PCA based analysis, the ID derived via TwoNN is substantially small. The paper stated that the learned representation fall in a distinct curved manifold rather than linear subspace. This might be biased since the TwoNN is non-linear analysis tool naturally.

3.In Table 5, the error rates of the model over training sets with full, long and short sequences are similar. This seems counterintuitive especially when training on short sequences. As we know, Transformer is an order-independent learning structure. For sequential learning, we need to add explicit position information (such as position embedding in Standard Transformer used in this paper). Training on short sequences will prevent its generalization ability on longer sequences. However, from Table 5, there is no significant difference between error rate especially for SST2 with diverse lengths.

4. What kind of distance is used in TwoNN for ID computation?

5. Missed related references
1) Anti-oversmoothing in deep vision transformers via the fourier domain analysis: From theory to practice. ICLR, 2022.
2) Revisiting over-smoothing in bert from the perspective of graph. ICLR, 2022.

---

### Official Review · Reviewer_rhBS · 2023-10-31

**Soundness:** 1 poor
**Presentation:** 1 poor
**Contribution:** 2 fair
**Rating:** 3
**Confidence:** 4

**Summary:**

This paper studies the intrinsic dimension of the dataset's hidden representation for text classification in transformer models. The analysis shows how the ID evolves through the layers and how it is affected by changes in embedding dimension and layer depth. The authors show that a higher intrinsic dimension of the last layer generally corresponds to higher classification accuracy and propose a strategy for selecting input data based on their analysis of the intrinsic dimension.

**Strengths:**

The analysis of the intrinsic dimension of the hidden representations in transformers trained on NLP tasks is an interesting and relevant topic that has only recently started to be addressed by some studies.

**Weaknesses:**

The experimental tests in support of some claims are not solid enough (see my concern regarding the ID vs. accuracy analysis).
Some parts contain technical flaws (see the concern about the PCA-ID computation), and the sentences are phrased in a way that is sometimes hard to follow.

**Questions:**

**Clarification about which feature space is used when computing the ID.**
The feature space in which the authors compute the ID needs to be better defined. The datasets analyzed consist of input sentences of variable length $l_i$, embedded in $l_i \times ED$ dimensional spaces by the transformer, where ED is the embedding dimension. The authors are likely applying the TwoNN in a space with a dimension equal to ED.

1. Where do the authors define the strategy to reduce sentences of different lengths to a common ambient space where the Euclidean distances can be computed?
In Sec. 3.2 the authors write that “every piece of input data is sequentially mapped in $n_l$ dimensional vector.
2. What do they mean by “piece of input data”: the tokens or the sentences? Is the meaning of hidden dimension $n_l$ different from that of embedding dimension ED? If not, why do they use different names?

**Correlation between ID and accuracy**
The authors claim that *an higher terminal feature ID correlates generally correlates with a lower classification error rate* (e.g., Abstract). However, they show incomplete/contradictory evidence about this aspect. In Sec. 3.3, the claim seems valid for AG and IMDB (but the evidence is not robust for SST2). \
More importantly, in Sec. 3.5 they show instead that the *ID of the last layer decreases* when the depth is increased, but they do not report the accuracy. Generally, increasing the transformer depth should also increase the accuracy:

3. Does the accuracy improve making the network deeper?

**Clarification about intrinsic dimension estimate with PCA.**
The ID estimate with PCA is usually done by looking for gaps in the eigenvalue spectrum of the covariance matrix. This approach is followed by Ansuini et al., which the authors cite and also widely reported in the technical literature (see e.g. [1-3]). In Sec. 3.4 the authors instead select an arbitrary threshold corresponding to 50% or 85% of the explained variance.

4. How do they justify their approach in this case?

[1] Little et al., Multiscale geometric methods for estimating intrinsic dimension. \
[2] Ceruti et al. Intrinsic Dimension Estimation: Relevant Techniques and a Benchmark Framework.\
[3] Kaslovsky and Meyer, Overcoming noise, avoiding curvature: Optimal scale selection for tangent plane recovery.

**Rationale of paragraphs 3.6-3.7**
I have some concerns about the rationale behind paragraphs 3.6-3.7. Decreasing the dataset size deteriorates the performance (especially on the SST2 dataset). The evidence that the sentence IDs do not change with the sentence length does not seem a useful criterion for the data selection.

**Minor suggestions**

a. The authors have trained the transformers but do not indicate the hyperparameter setup used (number of epochs, batch size, optimizer, learning rate). They should add this information to the manuscript or in the appendix.

b. The authors should add a link to the code so that others can reproduce the experiments described in the paper. This is now a *de facto* mandatory requirement in this field.

---

### Official Review · Reviewer_wsLx · 2023-11-01

**Soundness:** 2 fair
**Presentation:** 3 good
**Contribution:** 2 fair
**Rating:** 3
**Confidence:** 4

**Summary:**

This paper examines the intrinsic dimension (ID) of data in Transformer representations, as well as its trend across the Transformer layers, and the interplay between embedding dimension (ED), the intrinsic dimension (ID), and task performance.

**Strengths:**

originality: Although the intrinsic dimension (ID) is an established concept and the codes have been provided in prior work [1], this paper under review is the first to carefully examine ID (under this particular definition) in Transformers in the text classification setting.

clarity: The methodology and findings are clearly described.

[1] Intrinsic Dimension of Data Representations in Deep Neural Networks, NeurIPS 2019, https://github.com/ansuini/IntrinsicDimDeep

**Weaknesses:**

quality: a key underlying assumption is questionable, making the observations less convincing

Intrinsic dimension (ID) is defined in the following way: assuming a set of vectors “locally uniformly” lie on a d-dimensional subspace of the full D-dimensional vector space (d <= D), then the ratios of close pairwise distances follow the Pareto distribution parametrized by d. Then, the intrinsic dimension (ID) is the max likelihood estimation of d. (For a formal definition, see Section 2, paragraph “TwoNN Method” on page 3.)

Thus, the ID estimation is only meaningful if the following Assumption 1 holds:

Assumption 1: The ratios of close pairwise distances of Transformer representations (approximately) follow the Pareto distribution.

However, without further justification, the validity of Assumption 1 is questionable.

According to [2], a sufficient condition for Assumption 1 is that the representations are “locally uniform” in density, where "locally" means “within the range of the second neighbor for each data point”. It is unclear whether this sufficient condition holds for Transformer representations. Nevertheless, note that this is likely not a necessary condition, so there could be other ways to achieve Assumption 1. However, the authors did not check (or even explicitly mention) Assumption 1, which calls into question the validity of the method and the results.

Moreover, [1] states that for ID estimation for convolutional neural networks (CNNs) representations on image data,

“the estimated values remain very close to the ground truth ID, when this is smaller than ~20. For larger IDs and finite sample size, the approach moderately underestimates the correct value, especially if the density of data is non-uniform. Therefore, the values reported in the following figures, when larger ~20, should be considered as lower bounds.”

Note that in the current paper under review, most of the estimated IDs are greater than 20 (Figures 1-2, Tables 1-5). Can we trust the estimated IDs by this approach? Do we have evidence that the estimated values remain very close to the ground truth ID for a larger range of IDs (> 20) in the setting of this paper?



originality: it seems that the current paper under review mostly used the approach of [1] (which was for convolutional neural networks (CNNs) representations on image data), applying it to Transformers on text classification data. While some observations could potentially be interesting, the main technical contribution of the methodology should be primarily attributed to prior works.



[1] Intrinsic Dimension of Data Representations in Deep Neural Networks. NeurIPS 2019

[2] Estimating the intrinsic dimension of datasets by a minimal neighborhood information. Scientific reports, vol. 7, no. 1, p. 12140, 2017

**Questions:**

The reviewer is open to learning about new evidences or analyses which address the “quality” and “originality” points of the “Weaknesses” section above in this review.

---

### Official Review · Reviewer_K7iN · 2023-11-07

**Soundness:** 2 fair
**Presentation:** 2 fair
**Contribution:** 1 poor
**Rating:** 1
**Confidence:** 4

**Summary:**

This paper seems like a review paper or a "study" paper. They show a lot of experimental results to explore the intrinsic dimension calculated using the TwoNN technique. There is no novelty in this paper e.g. they don't propose a method.

**Strengths:**

None

**Weaknesses:**

This paper seems to be a study. And an incomplete one. It goes from related work where they describe the TwoNN method proposed by another paper, to experiment results. There is no technical section? It is also written kind of strangely.

**Questions:**

See weaknesses